# Adipose Tissue Remodeling in Obesity: An Overview of the Actions of Thyroid Hormones and Their Derivatives

**DOI:** 10.3390/ph16040572

**Published:** 2023-04-10

**Authors:** Giuseppe Petito, Federica Cioffi, Nunzia Magnacca, Pieter de Lange, Rosalba Senese, Antonia Lanni

**Affiliations:** 1Department of Environmental, Biological and Pharmaceutical Sciences and Technologies, University of Campania “L. Vanvitelli”, 81100 Caserta, Italy; 2Department of Sciences and Technologies, University of Sannio, 82100 Benevento, Italy

**Keywords:** adipose tissues, thyroid hormones, 3,5 diiodo-L-thyronine, browning, metabolic disease, obesity

## Abstract

Metabolic syndrome and obesity have become important health issues of epidemic proportions and are often the cause of related pathologies such as type 2 diabetes (T2DM), hypertension, and cardiovascular disease. Adipose tissues (ATs) are dynamic tissues that play crucial physiological roles in maintaining health and homeostasis. An ample body of evidence indicates that in some pathophysiological conditions, the aberrant remodeling of adipose tissue may provoke dysregulation in the production of various adipocytokines and metabolites, thus leading to disorders in metabolic organs. Thyroid hormones (THs) and some of their derivatives, such as 3,5-diiodo-l-thyronine (T2), exert numerous functions in a variety of tissues, including adipose tissues. It is known that they can improve serum lipid profiles and reduce fat accumulation. The thyroid hormone acts on the brown and/or white adipose tissues to induce uncoupled respiration through the induction of the uncoupling protein 1 (UCP1) to generate heat. Multitudinous investigations suggest that 3,3′,5-triiodothyronine (T3) induces the recruitment of brown adipocytes in white adipose depots, causing the activation of a process known as “browning”. Moreover, in vivo studies on adipose tissues show that T2, in addition to activating brown adipose tissue (BAT) thermogenesis, may further promote the browning of white adipose tissue (WAT), and affect adipocyte morphology, tissue vascularization, and the adipose inflammatory state in rats receiving a high-fat diet (HFD). In this review, we summarize the mechanism by which THs and thyroid hormone derivatives mediate adipose tissue activity and remodeling, thus providing noteworthy perspectives on their efficacy as therapeutic agents to counteract such morbidities as obesity, hypercholesterolemia, hypertriglyceridemia, and insulin resistance.

## 1. Introduction

The prevalence of obesity is well acknowledged worldwide as a major health issue. Due to changes in food composition and sedentary lifestyles in Western societies, obesity has reached epidemic proportions [1]. Excessive weight gain causes an increased risk of several diseases, mostly cardiovascular diseases (CVDs), type 2 diabetes (T2DM), non-alcoholic fatty liver disease (NAFLD), and cancer [2,3,4]. White adipose tissue (WAT) is the main storage site for excess calorie intake [5]. It ensures the survival of an organism during long periods of fasting [6]. It is also an organ that is able to respond rapidly and dynamically to nutrient deprivation and excess through adipocyte hypertrophy and hyperplasia [7]. In obese individuals, WAT exhibits reduced angiogenesis, excessive production of extracellular matrix (ECM), increased infiltration of immune cells, and consequent pro-inflammatory responses [8]. The remodeling of WAT is a continuous process that is pathologically accelerated in the state of obesity. In mammals, including humans, another major type of adipose tissue (AT), namely brown adipose tissue (BAT), is present in addition to WAT. Unlike white fat, the BAT is a specialized tissue for non-shivering thermogenesis (NST) to dissipate energy as heat, thereby playing a key role in the energetic homeostasis of the entire body. Thyroid hormones (THs) and their derivatives exert numerous functions in many tissues, including the activation and remodeling of AT. THs contribute significantly to brown adipocyte thermogenesis, and in white adipose depots, THs are also able to induce brown adipocyte recruitment, known as “browning”. The browning of WAT is certainly worthy of in-depth analysis to promote targeted therapeutic weight loss [9,10]. Herein, we summarize our current knowledge regarding the mechanism by which THs and their derivatives mediate AT activity and remodeling. This review provides relevant information on their potential use as therapeutic agents for the treatment of obesity and associated diseases.

## 2. Types of Adipose Tissue

AT plays a critical role in regulating body metabolism and homeostasis. Unlike other organs and tissues, AT can expand to accommodate excess energy in the form of accumulated lipids, distinguishing it from other organs and tissues [11]. AT is divided into two major types, WAT and BAT. Most of the WAT deposits are widely characterized as visceral (vWAT) or subcutaneous (sWAT) [12]. vWAT is further subdivided into mesenteric, omental, perirenal, and peritoneal deposits [13] (Figure 1). Alternatively, BAT is a thermogenic tissue specialized in the conversion of lipids into heat. It is indeed perfused by an extensive network of blood capillaries and highly innervated by noradrenergic fibers [14]. In addition to WAT and BAT, a third type of adipocyte has been described, termed “beige”, or “brite” (brown-in-white) [15]. In this tissue, white adipocytes show high plasticity and possess multiple similarities to brown adipocytes. Beige adipocytes are fundamental in weight control, energy balance regulation, and amelioration of glucose and lipid metabolism [9]. A growing body of evidence indicates that pathophysiological conditions such as obesity and aberrant remodeling of AT can induce dysregulation in the production of various adipocytokines, hormones, and metabolites, thus resulting in metabolic disorders [16].

## 3. WAT

WAT comprises adipocytes, which are bound by loose, vascularized, and innervated connective tissue. In white adipocytes, a large “unilocular” lipid droplet occupies more than 90% of the cell volume. Additionally, a thin layer of cytoplasm containing other organelles is present [17]. WAT plays an endocrine role as well as exerting a metabolic function. The metabolic functions include lipogenesis, fatty acid oxidation, and lipolysis, while adipokines are produced by the endocrine system. During fasting periods, WAT supplies fuel to the organism by storing and releasing highly energetic molecules, particularly fatty acids. The balance between lipid synthesis and fatty acid oxidation, as well as fatty acid release, is essential for adipocyte function [18]. Adipocytes secrete a variety of mediators, including exosomes, miRNAs, lipids, inflammatory cytokines, and peptide hormones [19,20]. However, numerous investigations have been conducted as regards the secretion of hormones by WAT. Among these hormones are leptin, adiponectin, and resistin which regulate food intake, the reproductive axis, insulin sensitivity, and immune responses. It has been shown that dysregulation of any of these hormones can lead to systemic metabolic dysfunction, as well as chronic metabolic diseases and several cancers [21].

## 4. Hypertrophic and Hyperplasic WAT Expansion

Metabolic syndrome/obesity is currently considered a burdensome and important health issue that may prompt the occurrence of related pathologies such as T2DM, hypertension, and cardiovascular disease (CVD). It is widely recognized that metabolic disorders are multifactorial pathologies in which nutrient excesses are closely related to the activation of the innate immune system in most organs responsible for maintaining energy balance [22]. In the presence of a positive energy balance, there is an increase in demand for lipid storage, which can be accommodated either by increasing adipocyte size (hypertrophy) or by increasing the number of adipocytes (hyperplasia) [23]. Adipocyte hypertrophy and hyperplasia are regulated by environmental and genetic factors [24]. According to some research, obesity is associated with an increase in dead adipocytes, which impair AT function and promote subsequent inflammation. In hypertrophic adipocytes, pro-inflammatory cytokines, including tumor necrosis factor α (TNF-α), interleukin 6 (IL-6), interleukin 8 (IL-8), and monocyte chemoattractant protein-1 (MCP-1), are expressed and secreted [25]. An increase in pro-inflammatory cytokines leads to insulin resistance and inflammation of AT due to the recruitment of macrophages and T cells [26,27]. Moreover, increased adipose mass is associated with hypoxia [28]. Activation of hypoxia-inducible factor 1α (HIF-1α), a key transcription factor mediating hypoxic responses, accelerates adipose tissue fibrosis. Alternately, hyperplasia is observed. The remodeling of AT involves reversible changes in the composition of immune cells and the size of adipocytes, altering numerous AT functions. Adipose tissue-activated macrophages (ATMs) polarize in response to changes in their environment, forming M1 and M2 macrophage phenotypes [29]. AT inflammation and insulin resistance (IR) in the whole body are initiated and maintained by classically activated M1 macrophages [30]. A large number of inflammatory cytokines, including IL-6, interleukin 1β (IL-1β), and MCP-1, are produced by macrophages that infiltrate the target organ under the condition of obesity. These cytokines negatively influence the transmission of insulin signals and increase the development of chronic inflammation and IR [31,32,33]. The M2 phenotype is, however, mainly responsible for anti-inflammatory responses and maintaining tissue homeostasis [34]. In general, M1 macrophages are mainly induced by Th1 signaling. On the other hand, M2 macrophages are induced by Th2 signaling and release arginase-1, interleukin 10 (IL-10), interleukin 4 (IL-4), interleukin 13 (IL-13), ornithine, and polyamines, promoting proliferation, tissue repair, and immune tolerance [35]. It is estimated that more than 90% of ATMs in healthy AT are of the M2 phenotype containing anti-inflammatory properties [36].

## 5. BAT

In the last two decades, positron emission tomography scans have revealed that metabolically active BAT exists in specific regions of adult humans. The amount of this tissue correlates positively with resting metabolic rate and negatively with body mass index [37]. BAT is characterized by multilocular lipid droplets; high mitochondrial density, which contributes to its coloration; and high expression of uncoupling protein 1 (UCP1). BAT is mainly located in the interscapular (iBAT) and subscapular (sBAT) regions of adult mice. In addition, small deposits of BAT are found deep between the scapula and the head (cervical BAT (cBAT)), around the aorta in the thoracic cavity (mediastinal BAT (mBAT)), and around the kidney (perirenal BAT (pBAT)). iBAT is the most widely used depot for the study of BAT function in mice [38]. The main role of BAT is to maintain a constant body temperature by generating heat. Unlike WAT, BAT is involved in non-shivering thermogenesis, a process that produces heat either through mitochondrial respiration uncoupling dependent on UCP1 or through a UCP1-independent mechanism [39]. Compared to white adipocytes, which contain a single lipid droplet, brown adipocytes contain many smaller droplets and a substantial number of mitochondria. In addition, brown fat contains more capillaries than white fat; capillaries supply oxygen and nutrients to the tissues and distribute heat throughout the body. BAT cells derive from a muscle-like Myf5+ cell line. PR domain containing 16 (PRDM16) controls the determination between brown fat and muscle between days 9 and 12 of pregnancy [40,41]. As previously mentioned, heat production is due to a UCP1 protein found almost exclusively in brown adipocytes. Additionally, diet-induced obesity is associated with decreased UCP1 expression and BAT thermogenesis [42]. During its active phase, BAT is able to absorb large quantities of lipids, glucose, and lactate from circulation, thereby affecting triglyceride levels as well as glucose concentrations in the blood, playing a fundamental role in metabolic homeostasis [43]. However, recent data suggest that BAT may play an endocrine role through the release of endocrine factors. Under conditions of thermogenic activation, brown fat releases several endocrine signaling molecules [44]. Although the endocrine role of BAT is still unknown, accumulating evidence indicates that BAT releases factors that act with both autocrine and paracrine action. These include vascular endothelial growth factor-A (VEGF-A), which probably promotes angiogenesis in response to sympathetic nervous system activation, insulin-like growth factor-I (IGF-I), and fibroblast growth factor-2 (FGF2) promoting an increase in the density of brown adipocyte precursor cells [45]. In BAT depots, thyroxine can be converted into 3,3′,5-triiodothyronine (T3) through the presence of type II 5′-deiodinase (BAT 5′D-II). Locally generated T3 contributes to the intracellular pathways of thermogenic activation of brown adipocytes [46]. Compared to white fat, BAT is less susceptible to developing local inflammation in response to obesity. Fitzgibbons TP et al. demonstrated that the BAT of obese mice exhibits significantly lower macrophage infiltration and immune cell-enriched mRNA expression than WAT, suggesting that this tissue “resists” obesity-induced inflammation [47]. Research studies have shown that mice fed a long-term HFD had elevated levels of inflammation markers such as TNF-α and EGF-like module-containing mucin-like hormone receptor-like 1 (F4/80). However, the increased levels of pro-inflammatory cytokines were mainly associated with the presence and activity of infiltrating pro-inflammatory immune cells [48,49].

## 6. Beige Adipose Tissue (BeAT)

In addition to WAT and BAT, a third type of adipocyte has been described, termed “beige”, or “brite” (brown-in-white). Similar to brown adipocytes, these are multilocular cells with moderate mitochondrial content and inducible expression of UCP1 [50]. The beige adipocytes arise from Myf 5-negative (Myf 5-) precursors [15,51]. It has been shown that some beige adipocytes express myosin-heavy chain 11 (Myh11), a selective marker of smooth muscles [51]. Recent findings have suggested that beige cells could originate from the transdifferentiation of white fat cells [52,53]. However, it is not clear whether this conversion involves a cell type with this specific predisposition. Therefore, several studies have been conducted to demonstrate that different WAT depots bear different browning capacities that most likely resort to alternative mechanisms to originate beige cells [54,55,56]. As observed in obesity, the infiltration of immune cells in the sWAT alters the ability of precursor cells to differentiate into active beige adipocytes and creates a deleterious inflammatory microenvironment involving TNF-α, interferon-c (IFN-c), and interleukin 17 (IL-17) [57]. Similarly to BAT, BeAT affects the entire body’s energy balance, and numerous investigations are underway to develop novel treatments for obesity and related complications. Loss of BeAT leads to obesity susceptibility [58]; indeed, when exposed to cold or activated by beta-adrenergic receptors (β-ARs), beige adipocytes can be detected in WAT. This phenomenon is known as the “browning of WAT” [59,60]. As indicated previously, BAT plays a key role in thermogenesis, contributing to energy consumption and the prevention of obesity. In addition to classic BAT activation for treating obesity and T2DM, the recruitment of beige adipocytes has received much attention in recent years. This could represent a novel therapeutic target for obesity and T2DM. Beige adipocytes are also crucial in weight control, energy balance regulation, and amelioration of glucose and lipid metabolism. External stimuli (cold exposure, β-adrenergic agonists, etc.) accelerate beige adipocyte recruitment by WAT, resulting in increased energy consumption and thermogenesis. The consumption of glucose and lipids indirectly improves glucose tolerance, insulin sensitivity, and beta-cell function [61,62].

## 7. Pink Adipose Tissue

Pink adipocytes are an alternative class of adipocytes that has recently generated interest in the scientific world. Pink adipocytes are alveolar epithelial cells of the mammary gland that produce and secrete milk for the nourishment of the pups [63]. It has been hypothesized that they derive from subcutaneous white adipocytes that have transdifferentiated [64]. It has also been observed that post-lactational pink adipocytes may trans-differentiate into brown adipocytes [65].

## 8. Thyroid Hormones and AT

### 8.1. Thyroid Hormone Biosynthesis and Actions: The Effect of 3,5,3′-Triiodo-L-thyronine (T3) on AT

The primary product of the thyroid is 3,5,3′,5′-tetraiodo-L-thyronine (T4), which is synthesized at three “hormonogenic sites” on the thyroglobulin chain. Despite exhibiting certain biological activities, T4 is a precursor to the active hormone, T3 [66]. In target tissues, T4 is also converted to T3 by type I 5′-deiodinase (D1) and type II 5′-deiodinase (D2). Further deiodination of circulating T4 and T3 produces TH derivatives that bind poorly to thyroid receptors (TRs). From T4, type III 5-deiodinase (D3) generates 3,3′,5′-triiodothyronine (reverse T3) and 3,3′-diiodo-l-thyronine (3,3′-T2). Together, the deiodination processes of T3 and rT3 give rise to different diiodothyronines and monoiodothyronines that are present in trace concentrations in the sera [67]. THs are hormones that affect almost all cells of the human body. Generally, THs increase metabolic rate and thus thermogenesis by binding their intranuclear receptor. Increased metabolic rate results in a greater consumption of oxygen and energy [68]. The effects of THs on target cells are exerted by different pathways, which can be subdivided into “genomic” and “nongenomic” actions [69]. The genomic actions of THs are initiated in the cell nucleus via TRs. T3 receptors are ligand-activated transcription factors. TRs can interact with thyroid-hormone response elements (TREs) as protein dimers, heterodimerizing with another member of the nuclear receptor family or with retinoic acid receptors (RXRs), or self-homodimerizing. In addition to the conventional effects of transcriptional regulation via TRs, THs exhibit a remarkably rapid action on cells which may be initiated outside the nucleus and involve a variety of signal transduction pathways (non-genomic action) [69]. The sites of nongenomic actions are distributed throughout various cellular compartments, including the plasma membrane, cytoplasm, cytoskeleton, and subcellular organelles (such as mitochondria) [70,71,72,73]. Membrane receptors, consisting of specific integrin αv/β3 receptors, have been identified and found to mediate actions at multiple sites [74,75].

### 8.2. The Effect of 3,5,3′-Triiodo-L-thyronine (T3) on AT

THs exert pleiotropic actions, and AT is an important target of these hormones [76]. Panveloscki-Costa et al. have demonstrated the beneficial effects of T3 treatment of obese rats on the improvement of insulin sensitivity and on the negative modulation of inflammatory state in epididymal and mesenteric AT. Therefore, this study showed that T3 treatment reduces adiposity and increases the lean mass of obese rats. T3 also acts as an immunomodulatory agent reducing the content of inflammatory cytokines in the AT and promoting a phenotypic switch in AT macrophage polarization [77]. In an alloxan-induced diabetic rat model, T3 treatment (1.5 µg per 100 g BW) reduced serum TNF-α and epidydimal WAT (eWAT) expression of IL-6 and TNF-α. Additionally, treatment with T3 decreased serum levels of chemokine (C-C motif) ligand 2 (Ccl2) and F4/80 and expression levels of cluster of differentiation 68 (CD68) in eWAT. This condition leads to a reduction in immune cell infiltration [78]. As a thermogenic hormone, T3 is essential for a full metabolic response of BAT under maximal demands [79,80,81]. THs can stimulate BAT directly, through TRs expressed in brown adipocytes, and indirectly, through TRs expressed in hypothalamic neurons. THs act on brown adipocyte thermogenesis by increasing the stimulatory action of norepinephrine (NE), as well as enhancing the cyclic adenosine monophosphate (cAMP)-mediated acute rise in UCP1 gene expression (Figure 2) [82]. Recently, studies have revealed that thyroid hormones can also induce facultative thermogenesis through central mechanisms, as central hyperthyroidism is able to directly activate BAT in a manner dependent on AMP-activated protein kinase (AMPK) and induces browning in mice [83,84]. In brown adipose tissue, T3 stimulates thermogenesis by induction metabolic inefficiency through the activation of the mitochondrial UCP1 [80,85]. Similar to the liver or pituitary, BAT exhibits a high number of TRs [86]. There are two TR genes, thyroid hormone receptor α (TRα) and thyroid hormone receptor β (TRβ), which are differentially expressed during development and in adult tissues [87,88]. TRα has three splice products. TRα1 is located primarily in the brain, heart, and skeletal muscles and binds to T3. TRα2 and TRα3 splice products are non-T3-binding with several truncated variants [87]. TRβ has three major T3-binding splice products: TRβ1 is widely expressed; TRβ2 is mainly expressed in the brain, retina, and inner ear; and TRβ3 is widely expressed in the kidney, liver, and lung [87,89]. Both isoforms are required for adequate adaptive thermogenesis in BAT. T3-regulated UCP1 mRNA expression is mediated by TRβ, while TRα1 maintains brown adipocyte adrenergic responsiveness [90]. Mice with global deletion of TRα1, TRβ, or TRα isoforms showed cold intolerance associated with impaired BAT thermogenesis [86]. In brown adipocytes, DIO2 regulates local T3 levels. In mature brown adipocytes, D2-expressing cells produce high levels of T3 and activate thyroid hormone receptors [91]. The lack of adipose DIO2 causes abnormal lipid metabolism in BAT that subsequently leads to cold intolerance [92]. According to Martinez-Lopez et al., cold exposure at 4 °C induces lipophagy and mitophagy in BAT, suggesting that autophagy is required for adaptive thermogenesis [93,94]. Based on these findings, other researchers investigated whether T3 has a cell-autonomous role in BAT activation by examining autophagy, mitochondrial turnover, fatty acid metabolism, and mitochondrial respiration. The results suggest that T3 increases mitochondrial autophagy (mitophagy) and biogenesis to maintain mitochondrial quality control (MQC) [95]. Apart from regulating mature brown adipocytes’ thermogenic capacity directly or through the sympathetic nervous system (SNS), T3 can stimulate the hyperplastic growth of iBAT [96].

### 8.3. The Effect of 3,5,3′-Triiodo-L-thyronine (T3) on “Browning”

Recent research by Shengnan Liu et al. has demonstrated that T3 can promote adipocyte progenitor cell (APC) proliferation in the iBAT depot of mice. Considering that TRα mediates the T3 effect on APC proliferation in the iBAT depot, further analysis suggests that T3 promotes cell state transition and cell cycle progression via c-Myc in APCs [97]. Such effects of THs on BAT are well known; however, an alternative mechanism, the so-called “browning” of WAT, has been acknowledged as effectively supporting THs in energy expenditure. In an in vitro model of differentiated human multipotent adipose-derived stem cells (hMADSs), T3 treatment induced UCP-1 expression and mitochondrial biogenesis accompanied by the induction of PGC-1 (peroxisome proliferator-activated receptor-γ coactivator-1α) and NRF1 (nuclear respiratory factor 1). Such impacts of T3 on UCP-1 induction were dependent on TRs [98]. Moreover, in obese individuals, a reduction of the browning process in WAT was observed. Matesanz et al. demonstrated that the expression of the MAPK kinase 6 (MKK6) is increased in the WAT of obese individuals and reported that in knockout animals, the deletion of MKK6 increases T3-stimulated UCP1 expression in adipocytes, thereby enhancing their thermogenic capacity [99]. Another study by Miriane de Oliveira et al. showed that, in addition to improving UCP1 expression, T3 treatment improved lipid profile, oxidative stress, and DNA damage in human subcutaneous preadipocytes [100]. Activating BAT or “browning” of the WAT is therefore considered a promising therapeutic approach for treating obesity and metabolic disorders. In addition, very recently, Ma et al. have demonstrated that systemic administration of T3 affects both inguinal white adipose tissue (iWAT) and whole-body metabolism. They showed that TRβ is the major TR isoform that mediates the T3 action on multiple metabolic pathways in iWAT, including glucose uptake and usage, de novo fatty acid synthesis, and both UCP1-dependent and -independent thermogenesis [101].

## 9. 3,5-Diiodo-L-thyronine (T2) and Its Multiple Biological Effects on AT

### 9.1. 3,5-Diiodo-L-thyronine (T2), a Thyroid Hormone Derivative with Potent Metabolic Effects

Recently, evidence has emerged that some TH metabolites, previously considered inactive products of thyroid hormone metabolism, possess biological activities which include 3,5-T2, a compound that has been a focal point of our previous studies [102,103]. This metabolite manifests some effects of TH within one hour of administration, and mitochondria are considered a direct target of 3,5-T2 [104,105]. It has been reported that T2 has T3-like effects in the absence of thyrotoxic side effects, at least when used at low concentrations [106,107,108,109]. According to several studies, 3,5-T2 exerts significant biological effects in a variety of tissues, such as the liver, skeletal muscle, heart, and AT [102,110,111]. In hypothyroid rats, the administration of 3,5-T2 increased the resting metabolic rate [112], cold tolerance [113], and the ability to use lipids as metabolic substrates [114]. It has also been observed that chronic administration of 3,5-T2 to rats fed an HFD prevented body weight gain, liver adiposity, hypercholesterolemia, and hypertriglyceridemia, while concomitantly preserving muscle glucose uptake and insulin sensitivity [115,116,117,118]. In addition, in vivo studies show that 3,5-T2 exerts metabolically favorable effects on AT.

### 9.2. The Effect of 3,5-Diiodo-L-thyronine (T2) on BAT

Similarly to T3, 3,5-T2 affects BAT thermogenesis. It improves survival in the cold of hypothyroid rats, increases the oxidative potential of the cell directly binding to cytochrome c oxidase (COX), and induces an increase in mitochondrial biogenesis [119]. A study conducted by Lombardi et al. found that the intraperitoneal administration of 3,5-T2 to hypothyroid rats housed at thermoneutrality reversed the “white-like” appearance of brown adipocytes in such animals and increased the proportion of multilocular versus unilocular cells in such animals. In addition, 3,5-T2 decreases the diameter of lipid droplets (LDs) and increases mitochondrial content, indicating activation of the BAT [120]. 3,5-T2 also increases the cellular number of nerve fibers, suggesting that such a part of the thermogenic effect induced by this iodothyronine in BAT is due to sympathetic nervous system (SNS) activation. Moreover, in T2-treated animals, BAT vascularization was higher due to sympathetic activation, since adrenergic stimulation induces VEGF production [120,121].

### 9.3. The Effect of 3,5-Diiodo-L-thyronine (T2) on “Browning”

Interestingly, it has been demonstrated that 3,5-T2 can induce browning sWAT of rats housed at thermoneutrality [122]. The ability of 3,5-T2 to affect thermogenesis may also be related to changes in adipocyte morphology and functionality. A browning process has also been reported in a section of anterior sWAT of HFD-T2 rats in which several white adipocytes changed their phenotype. As a result of this transformation, the adipocyte acquires a multilocular phenotype as opposed to a conventional unilocular phenotype, displaying immunoreactivity for UCP1. This process involves different pathways, including microRNAs (e.g., miR-133a and miR-196a) and irisin [122].

### 9.4. The Effect of 3,5-Diiodo-L-thyronine (T2) on vWAT

Changes in AT mass and adipocyte volume are known to provoke broad metabolic consequences [123]. Recently, the effects of 3,5-T2 on vWAT of HFD rats have also been studied by proteomic analysis. Silvestri et al. demonstrated that 3,5-T2 promoted visceral adipose lipolysis through hormone-sensitive lipase (HSL) activation when administered simultaneously to rats treated with HFD (within 1 day after administration), while long-term treatment with 3,5-T2 affected adipocyte morphology (measurable after only 2 weeks and persistent to treatment), tissue vascularization, and the protein profile. In fact, 4 weeks of 3,5-T2 administration prevented HFD-induced hypertrophy and improved vVAT vascularization, suggesting that this iodothyronine may have proangiogenic properties contributing to its insulin-sensitizing properties [124]. Even more recently, an anti-inflammatory effect exerted by 3,5-T2 on vWAT of rats fed a long-lasting HFD (14 weeks) has been shown by Petito et al. Furthermore, they demonstrated that 3,5-T2 was able to induce a switch from M1 macrophages to M2 macrophages. In addition, the decrease in cluster of differentiation 45 (CD45) and cluster of differentiation 3 (CD3) expression levels and the increase in forkhead box P3 (Foxp3) levels indicate that 3,5-T2 suppresses lymphocyte recruitment. This study also showed that in HFD-T2 rats, the serum levels of irisin were increased, suggesting that this myokine could be a mechanism by which 3,5-T2 affects the inflammatory state. Additionally, to the best of our knowledge, this study reports for the first time that 3,5-T2 administration reduces the hypoxic environment induced by HFD and counteracts the DNA damage induced by oxidative stress occurring in overweight animals [125]. Most studies on 3,5-T2 effects have been conducted on animal models; therefore, the physiological effects of this metabolite on humans are still unclear, in particular, as regards the potential benefits in terms of obesity and related diseases based on the limited number of experiments performed [126]. Overall, considering the potentially beneficial effects, these results could support further studies to demonstrate the efficacy of 3,5-T2 as a therapeutic agent. The figure below illustrates a schematic representation of the effects exerted by 3,5-T2 on AT (Figure 3).

## 10. Thyroid Hormone Metabolites and Synthetic Analogs That Act on Adipose Tissue

Other natural metabolites that exert action on adipose tissue are thyronamines (TAMs). The TAMs are natural TH hormone derivatives without the carboxyl group on the alanine side chain. Nine TAMs have been described, with differences in iodine atom placement or number, but only 3-iodothyronamine (3-T1AM) and thyronamine (T0AM) have been identified in vivo [127]. 3-T1AM is found in T3 target tissues and the thyroid. However, the physiological or pathological significance of such tissues is still unknown [128]. T1AM administration in vivo has significant transcriptional effects, evident specifically in AT rather than in the liver. These effects may contribute to a reduction in fat mass and an increase in lipid metabolism [129]. Using brown adipocytes (BAs) isolated from rat BAT stromal fraction, Manuela Gencarelli et al. found that treating the cells with T1AM (M+T1AM) decreased cell lipid content, activated lipolysis, and shifted the cells into a catabolic state. According to these findings, BA long-term exposure to T1AM may ameliorate IR and obesity-related clinical conditions [130]. The subsequent oxidative deamination of iodothyronamines leads to the formation of iodothyroacetic acid derivatives [131]. Some of these metabolites, such as triiodothyroacetic acid (Triac) and tetraidothyroacetic acid (Tetrac), have been found to exert biological effects. Triac has a higher affinity for TRβ1 than T3 in various cell types including brown adipocytes [132]. An investigation performed on rats revealed that the T3 metabolite triiodothyracetic acid, at low doses, induced ectopic expression of UCP1 in abdominal WAT [133]. In addition to the naturally occurring metabolites, there are TH analogs that affect specific tissues by binding to the TR isoform in a specific manner. The most studied analog of THs that exhibits the beneficial metabolic properties of T3 is 3,5-dimethyl-4[(40-hydroxy-30-isopropylbenzyl)-phenoxy] acetic acid (GC-1). GC-1 has a high affinity for TRs and is selective in the binding and activation of TRβ over TRα [134]. Many studies have demonstrated the beneficial effects, mainly on dyslipidemia and obesity, with no unfavorable effects on the heart [135,136]. It has been shown that chronic administration of GC-1 to ob/ob mice resulted in marked browning of the subcutaneous WAT [137]. Noteworthy are the metabolic effects of GC-1 and mediation by WAT browning rather than an increase in BAT function, as revealed by the reduced expression of the UCP1 gene and the UCP1 protein [138]. Additionally, Lin et al. showed that chronic administration of GC-1 to obese mice markedly increased the browning of sWAT [138]. To date, little is known regarding the physiological perspective. Further research is required to comprehend the effect of such metabolites on AT physiology.

## 11. Conclusions

The remodeling of AT is a complex but well-orchestrated mechanism that allows adaptation to external environmental changes. A deeper understanding is required to better understand the remodeling of AT towards the development of therapeutic approaches in obesity-induced metabolic disorders. The THs and certain derivatives have been found to influence relevant metabolic/physiological pathways in AT. In addition, in the last decades, numerous studies have highlighted the positive effect of such compounds on the etiology and progression of obesity-linked metabolic disorders. Our investigations offer insight into their potential use as therapeutic agents to counteract diseases such as obesity, hypercholesterolemia, hypertriglyceridemia, and IR.

## Figures and Tables

**Figure 1 pharmaceuticals-16-00572-f001:**
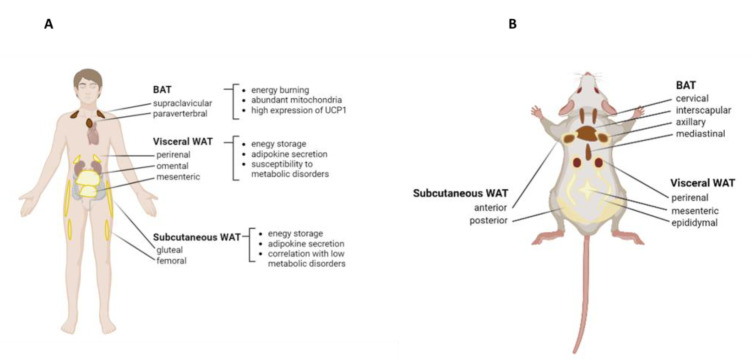
**Adipose tissue distribution.** (**A**) Distribution of WAT and BAT in humans. Both vWAT and sWAT possess abilities to store energy and secrete various adipokines. The sWAT is distributed throughout the body under the skin, while the vWAT surrounds the intra-abdominal organs. Located around the shoulders and ribs, BAT contributes to heat generation through the expression of UCP-1. (**B**) As compared to adult humans, the BAT in adult mice is well developed and easy to observe. The gonadal WAT depots located around the ovaries and the testes are studied as a model of vWAT. The figure was created with Biorender.com.(https://www.biorender.com/ Accessed on 16 May 2022).

**Figure 2 pharmaceuticals-16-00572-f002:**
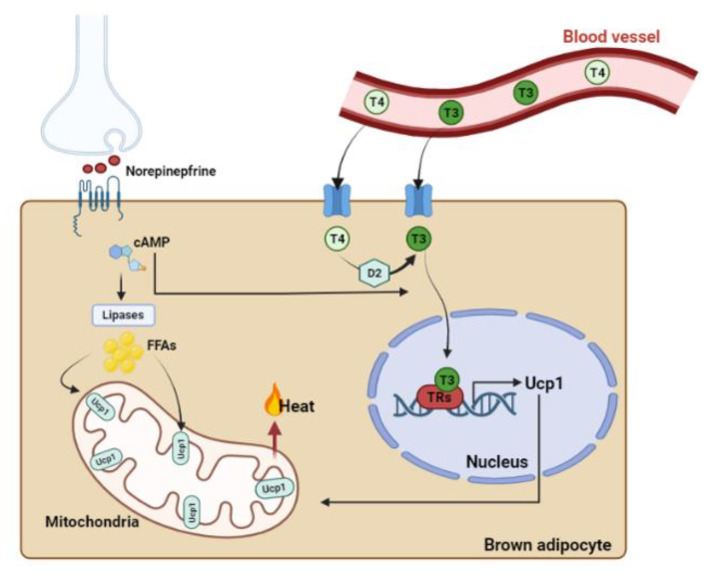
**Thermogenic control of brown adipocytes by UCP1.** Sympathetic neurons release synaptic norepinephrine that binds to β-adrenergic receptors, stimulating the production of cAMP by adenylate cyclase. The sympathetic signal activates transcription factors and coactivators involved in the regulation of DIO 2. Both adrenergic signaling and TRs regulate UCP1 expression. Lipases break down triglycerides into free fatty acids, which are then transported to mitochondria and activate UCP1. UCP1 uncouples ATP production from respiration, causing an increase in mitochondrial activity and generating heat. The figure was created with Biorender.com.

**Figure 3 pharmaceuticals-16-00572-f003:**
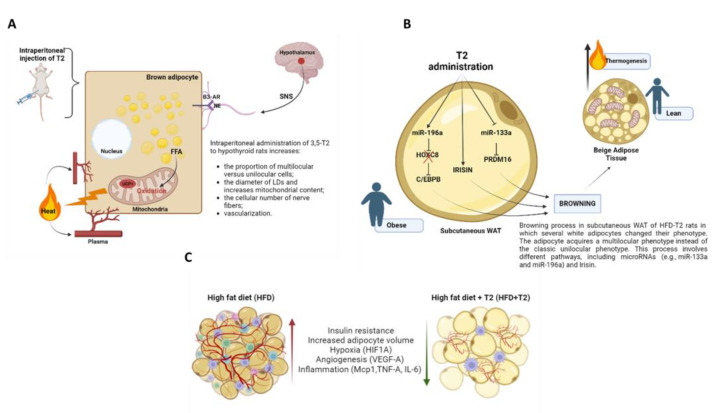
(**A**) In hypothyroid rats housed at thermoneutrality, 3,5-T2 increases multilocular versus unilocular cells, decreases LD diameter, and increases mitochondrial content, indicating BAT activation. Diiodothyronine also increases the cellular number of nerve fibers and vascularization, suggesting that part of the thermogenic effect induced by 3,5-T2 in BAT is due to SNS activation. (**B**) Browning process in a section of anterior sWAT of HFD-T2 rats in which several white adipocytes undergo phenotypic change. The adipocyte acquires a multilocular phenotype as an alternative to the conventional unilocular phenotype, with high levels of UCP1. This process involves different pathways, including microRNAs (e.g., miR-133a and miR-196a) and irisin. (**C**) Schematic representation of the effects exerted on adipocytes in vWAT by HFD for 14 weeks and by HFD for 14 weeks and 3,5-T2 administered daily during the last 4 weeks (HFD-T2). In vWAT from overweight rats, hypoxia induces the synthesis of several angiogenic factors (e.g., VEGF-A) and the expression of inflammatory cytokines (e.g., TNF-α, IL-6). Accordingly, a vicious circle is established in which the activation of angiogenesis first determines a further increase in adipocyte volume, thus enhancing an increase in the inflammatory state of the AT. In overweight rats treated with 3,5-T2, the inflammatory state is reverted. Diiodothyronine can reduce hypoxia, angiogenesis, and inflammatory agents. The figure was created with Biorender.com.

## Data Availability

Not applicable.

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
