# Peer review of "Adipose Tissue Remodeling in Obesity: An Overview of the Actions of Thyroid Hormones and Their Derivatives"

_pharmaceuticals, 2023, doi:10.3390/ph16040572_

Round 1

Reviewer 1 Report

General comments

The manuscript is for the most part well-written and clear. The review will be of interest to readers of Pharmaceuticals. I have no major concerns. One minor suggestion that the authors may consider that may help readability, is in places to separate the large sections of text into separate paragraphs. Each section currently consists of a single block of text, and for the longer sections dividing the text into paragraphs may help make the text less dense.  

Specific comments

Figures 1 and 3: The labels within the figure are rather small and hard to read. I would suggest making them slightly larger.

Author Response

Reviewer: The manuscript is for the most part well-written and clear. The review will be of interest to readers of Pharmaceuticals. I have no major concerns. One minor suggestion that the authors may consider that may help readability, is in places to separate the large sections of text into separate paragraphs. Each section currently consists of a single block of text, and for the longer sections dividing the text into paragraphs may help make the text less dense

Answer: We thank the Reviewer for his/her positive comment and careful review, which helped improve the quality of our manuscript. We separated the large sections of the text in more paragraphs. See lines 326, 327; 357; 405; 441-442; 458; 472; 481 of the revised manuscript. We hope that he/she appreciates this.

Reviewer: Figures 1 and 3: The labels within the figure are rather small and hard to read. I would suggest making them slightly larger.

Answer: We agree with the reviewer and the labels have been enlarged and the figures have been replaced. See lines: 165, 431, and 510 of the revised manuscript.

Reviewer 2 Report

The authors present a review describing the different types of adipose tissue focusing primary on white and brown adipose tissue. They describe some of the cellular and physiological differences between the tissues and the health risks associated with white adipose tissue. They then discuss the role of thyroid hormone and some of its derivatives in the remodeling of adipose tissue and the reduction of health risk factors, suggesting a potential therapeutic role for these compounds.

The review is well written and easy to understand, I only have a few minor recommendations or corrections for you consideration.

You discuss some differences between white and Brown fat white I presume have these names based on their physical appearance and mention some of the changes in the lipid structure but do not mention whether it is the cell organelles or lipids which change its appearance.

It is unclear how the second part of the paragraph for pink adipocytes, (lines 218-226 relate to the pink adipocytes.

Line 341 - I don't think "hypothyroid" is  a correct term but it conveys the idea.

Line 409 - TNF-A = I think you wanted to use the alpha symbol instead of "A"

Author Response

Reviewer: You discuss some differences between white and Brown fat white I presume have these names based on their physical appearance and mention some of the changes in the lipid structure but do not mention whether it is the cell organelles or lipids which change its appearance.

Answer: We thank the Reviewer for his/her positive comment and careful review, which helped improve the quality of our manuscript. The main morphological differences between brown and white adipose tissue are the following: the cytoplasm of the BAT occurs throughout the cell, and the nucleus can be found at the center of the cell, instead, the cytoplasm of the WAT occurs as a narrow rim, and the nucleus of the cell is pressed near the margin. Both white and brown adipose tissue contain lipid droplets within the cytoplasm. The BAT is multilocular, while, WAT is unilocular. Moreover, BAT consists of many mitochondria, while, WAT consists of a few mitochondria. BAT coloration is due to the fact that it is more vascularized and has a high content of mitochondria, which, in turn, have cytochromes, which are responsible for giving color.

We added more detailed description in the revised manuscript. See lines: 235-236.

Reviewer: It is unclear how the second part of the paragraph for pink adipocytes, (lines 218-226 relate to the pink adipocytes.

Answer: We would like to thank the reviewer for taking the time and effort to review the manuscript. The pink adipocytes paragraph has been edited to remove the ambiguous part about inflammation of mammary fat. See lines of the revised manuscript 314-315.

Reviewer: Line 341 - I don't think "hypothyroid" is a correct term but it conveys the idea.

Answer: We thanks the reviewer for the comment. In agreement with reviewer we rewrite sentence. See line 450 of the revised manuscript.

Reviewer: Line 409 - TNF-A = I think you wanted to use the alpha symbol instead of "A"

Answer: We thanks the reviewer for the comment. In agreement with reviewer we changed A with α. See line 525 of the revised manuscript.

Reviewer 3 Report

The review reads well and it is a comprehensive review of the topic. The standard of english is good and it is well referenced throughout. 

One minor change required in Figure legend 2 line 327 Ucp1 should be changed to UCP1 

The standard of figures are very good and relevant to the paper 

Author Response

Reviewer: One minor change required in Figure legend 2 line 327 Ucp1 should be changed to UCP1.

Answer: We thank the reviewer for his/her positive comment and careful review. In agreement with reviewer, we changed “ucp1” with UCP1. See line 435 of the revised manuscript.